# What Makes Effective Supervision in Latent Chain-of-Thought: An Information-Theoretic Analysis

Xinghao Chen [1 2]   Chak Tou Leong [2]   Wenjin Guo [1]   Jian Wang [2]   Wenjie Li [2]   Xiaoyu Shen [1]

## Abstract

Latent Chain-of-Thought (CoT) internalizes reasoning within continuous hidden states, offering a promising alternative to verbose discrete reasoning traces. However, robust latent reasoning remains difficult because outcome supervision provides weak learning signals and leaves latent trajectories prone to semantic drift. In this work, we analyze Latent CoT from an information-theoretic perspective and identify this failure as a dual collapse: gradient attenuation along the optimization path and representational drift in the latent space. We further decompose process supervision into two complementary dimensions: Trajectory Supervision, which injects dense stepwise reasoning signals, and Space Supervision, which preserves the semantic structure of the latent manifold. Our analysis shows that rigid geometric compression can collapse the reasoning space, whereas generative reconstruction provides a more flexible semantic anchor that better preserves information capacity. To measure these effects, we introduce the Unified Latent Probe (ULP), which quantifies the mutual information between latent trajectories and explicit reasoning steps. Experiments reveal a clear Information-Performance Binding: reasoning accuracy depends on the information fidelity preserved in the latent chain. These findings provide a principled framework for latent reasoning supervision and suggest shifting from geometric imitation toward mutual information maximization. Our code is available at this repository.

[1]Ningbo Institute of Digital Twin, Eastern Institute of Technology, Ningbo, China [2]Department of Computing, The Hong Kong Polytechnic University, Hong Kong, China. Correspondence to: Xiaoyu Shen <xyshen@eitech.edu.cn>.

*Proceedings of the 43[rd] International Conference on Machine Learning*, Seoul, South Korea. PMLR 306, 2026. Copyright 2026 by the author(s).

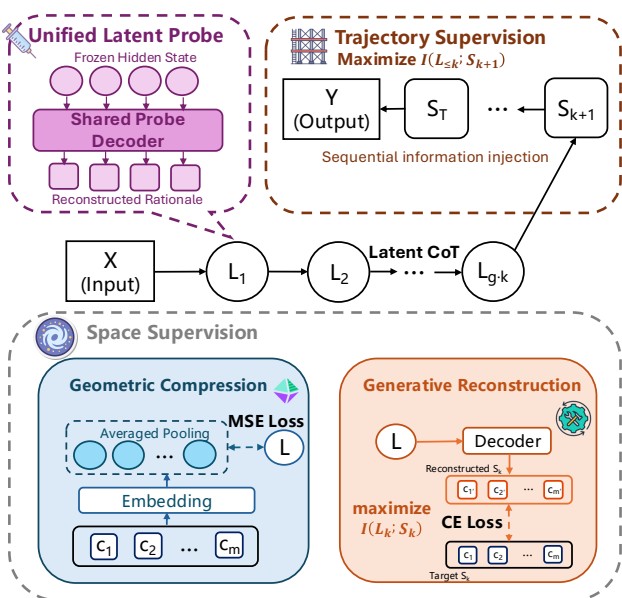

*Figure 1.* Overview of our information-theoretic framework for Latent CoT. Process supervision is decomposed into trajectory supervision for sequential information injection and space supervision for semantic anchoring, while ULP probes frozen latent states to quantify recoverable reasoning information.

## 1. Introduction

Large Language Models (LLMs) have achieved strong performance on complex reasoning tasks by generating explicit chain-of-thought (CoT) sequences (Wei et al., 2022; Guo et al., 2025; Chen et al., 2025b). However, representing reasoning as verbose sequences of natural language introduces intrinsic constraints. First, explicit CoT suffers from expressive redundancy: many tokens in a reasoning chain are syntactically necessary but functionally irrelevant, leading to inflated sequence lengths without proportional gains in reasoning quality (Feng et al., 2025; Li et al., 2025). Second, it imposes a semantic bottleneck: as abstract, continuous, and compositional reasoning processes cannot be faithfully represented within natural language space, inevitably causing information loss (Chen et al., 2025d;a; Sun et al., 2025). These limitations have motivated Latent Chain-of-Thought, which internalizes reasoning within continuous hidden states

rather than externalizing it as text (Hao et al., 2025; Shen et al., 2025). By removing the low-bandwidth constraint of discrete tokens, latent CoT enables a more expressive and computationally efficient form of internalized "System 2" reasoning (Deng et al., 2024; Geiping et al., 2025).

Despite its promise, achieving robust latent CoT remains challenging (Lindsey et al., 2025; Wei et al., 2025). A central difficulty lies in the representation shift from the discrete text space used in pre-training to high-dimensional continuous latent spaces, which requires models to adapt to a fundamentally different regime. Furthermore, this latent space is not directly interpretable, making it difficult to provide explicit supervision (Chen et al., 2025c; Tan et al., 2025; You et al., 2026). While a growing body of work has introduced additional supervision strategies to stabilize training (Hao et al., 2025; Shen et al., 2025; Butt et al., 2025; Deng et al., 2026; Li et al., 2026), the field lacks a principled framework for understanding how these signals govern the geometry and dynamics of the latent reasoning space.

To bridge this gap, we formalize Latent CoT through an information-theoretic lens. We begin by examining the most intuitive baseline: outcome supervision, which attempts to directly maximize the mutual information between latent reasoning steps and the final answer. Empirically, this approach performs no better than standard non-CoT baselines. We trace this failure to a dual collapse. From an optimization standpoint, it suffers from gradient attenuation; the sparse outcome signal decays so rapidly across the latent trajectory that credit assignment fails. From a representational standpoint, the latent manifold progressively drifts away from the semantic structure of the token space, indicating that *outcome supervision alone is insufficient to induce meaningful latent CoT steps*.

Given the dual collapse of outcome supervision, we review the fragmented landscape of existing Latent CoT research and identify a broader family of **process supervision** methods, which use explicit rationales to provide intermediate guidance beyond the final answer. We deconstruct process supervision into two conceptually distinct but interacting dimensions: *trajectory supervision* and *space supervision*. Figure 1 provides an overview of this framework.

Trajectory supervision addresses the temporal decay of information by providing dense, stepwise signals, training latent states to predict "suffixes" of explicit reasoning sequences. This acts as a mechanism for sequential information injection, effectively maximizing the mutual information between current latent steps and future reasoning trajectories. Space supervision, conversely, manages the structural integrity of the manifold itself by anchoring it to a known semantic domain. This is achieved either through geometric compression, which forces latent states into a rigid alignment with token embeddings, or generative reconstruction, which requires the states to remain decodable into natural language. Both approaches attempt to maximize mutual information between latent states and corresponding CoT steps, but with different inductive biases.

Our empirical analysis reveals a delicate interplay between these dimensions. Process supervision significantly stabilizes training. We observe that inserting new latent steps leads to increased gradient magnitudes, indicating active adaptation rather than attenuation. Furthermore, we find that optimizer resetting is crucial for inducing effective training signals at each stage, preventing interference between newly introduced latent variables and previously learned representations. The effectiveness of space supervision signals, however, is highly dependent on the nature of supervision applied. While geometric compression often introduces a "destructive prior" that collapses the reasoning manifold and reduces its information capacity, generative reconstruction acts as a flexible semantic tether. It preserves the intrinsic dimensionality of the latent space and serves as a calibration mechanism that mitigates semantic drift.

To quantify these effects, we introduce the **Unified Latent Probe (ULP)**, a variational reconstruction-based diagnostic framework that trains a shared conditional decoder to recover step-wise explicit reasoning from frozen latent hidden states extracted across models. By operationalizing mutual information as a reconstruction objective over a unified probing dataset, ULP provides a principled measure of how much structured reasoning is preserved and recoverable in continuous latent space. This reveals a clear information–performance binding: reasoning accuracy scales with the amount of information preserved in latent representations. Our contributions are as follows:

- Diagnosis of the Optimization Barrier: We demonstrate that the failure of outcome-supervised Latent CoT is an information-theoretic collapse, and that dense, sequential information injection (process supervision) is required to induce valid latent dynamics.

- The Unified Latent Probe (ULP): We design a rigorous diagnostic instrument that solves the "black box" verification problem by quantitatively linking representational fidelity (mutual information) to downstream reasoning performance.

- Generative Reconstruction over Geometric Compression: We establish that effective supervision must prioritize the preservation of information capacity via. e.g., generative reconstruction rather than rigid geometric compression.

## 2. Preliminaries and Problem Formulation

We formalize the problem of complex reasoning as learning a conditional distribution $p(y|x)$ that maps an input query $x \in \mathcal{X}$ to a final answer $y \in \mathcal{Y}$. For tasks requiring multi-step logic, learning this direct mapping reliably is difficult, as a single forward pass must implicitly capture all intermediate reasoning. Standard approaches address this by introducing an explicit intermediate rationale that decomposes the reasoning process into tractable steps.

### 2.1. Formulation of Reasoning Dynamics

**Explicit Chain-of-Thought.** In the standard paradigm, the rationale is a sequence of discrete tokens $C = (c_1, c_2, \ldots, c_N)$ drawn from a fixed vocabulary $\mathcal{V}$. The joint distribution factorizes as:

$$p(y, C|x) = p(y|C, x) \prod_{i=1}^{N} p(c_i | c_{<i}, x). \quad (1)$$

Projecting abstract reasoning onto discrete symbols acts as a *lossy compression* of the underlying thought process, fundamentally constrained by the finite vocabulary and the discrete nature of language.

**Latent Chain-of-Thought.** We investigate *Latent Chain-of-Thought*, where this scaffold is internalized into a trajectory of continuous hidden states $L = (L_1, L_2, \ldots, L_T)$. Each reasoning step $L_t$ consists of a sequence of $g$ continuous vectors, where $g$ denotes the latent granularity:

$$L_t = (h_{t,1}, \ldots, h_{t,g}) \in \mathbb{R}^{g \times d}, \quad (2)$$

where $d$ is the hidden dimension. The inference marginalizes over the latent trajectory:

$$p(y|x) = \int p(y|L, x) p(L|x) \, dL. \quad (3)$$

**The Representational Interface Gap.** The contrast between explicit and latent CoT can be viewed as a difference in the bandwidth of the intermediate reasoning interface. Let $R$ denote an idealized random variable representing the task-relevant reasoning state underlying a complete solution. For explicit CoT, this information must pass through $C \in \mathcal{V}^N$. Since $C$ consists of $N$ vocabulary tokens, its conditional entropy is upper-bounded by

$$H(C \mid x) \leq N \log_2 |\mathcal{V}|. \quad (4)$$

In practice, syntactic and semantic constraints of natural language make the empirical entropy of emitted rationales substantially lower. Equivalently, any information about the underlying reasoning process that is recoverable from the emitted rationale satisfies

$$I(R; C \mid x) \leq H(C \mid x) \leq N \log_2 |\mathcal{V}|. \quad (5)$$

If the rationale corresponds to $T$ conceptual reasoning steps with an average of $\bar{n}$ tokens per step ($N \approx T\bar{n}$), this gives

$$I(R; C \mid x) \lesssim T\bar{n} \log_2 |\mathcal{V}|. \quad (6)$$

Latent CoT instead exposes a sequence of continuous intermediate states $L = (L_1, \ldots, L_T)$, where each $L_t \in \mathbb{R}^{g \times d}$. For a simplified upper-bound comparison, we consider a finite-precision or noise-limited discretization $\tilde{L} = Q(L)$, with at most $B_{\text{eff}}$ reliable bits per dimension. Then the information recoverable from the latent interface is bounded by

$$I(R; \tilde{L} \mid x) \leq H(\tilde{L} \mid x) \leq T g d B_{\text{eff}}. \quad (7)$$

This suggests that latent reasoning states can encode richer non-linguistic computational features that are inevitably lost under explicit symbolic compression (Zhu et al., 2025a; Zou et al., 2025), since $dB_{\text{eff}}$ can substantially exceed $\log_2 |\mathcal{V}|$. High-dimensional latent states may carry task-relevant computation, but they may also encode noise, spurious correlations, or representations that drift away from the semantic structure learned during language pre-training.

### 2.2. Supervision Challenges in Latent CoT

The theoretical objective of latent Chain-of-Thought (CoT) reasoning is to maximize the mutual information (MI, Shannon 1948) between the latent trajectory $L$ and the target answer $y$, such that each latent state $L_t$ captures answer-relevant intermediate information. However, unlike discrete reasoning tokens, latent states have no observable ground truth, and the optimal latent trajectory remains inherently hidden. This makes supervision fundamentally challenging: optimizing solely for the final answer provides no direct guarantee that the learned latent states correspond to meaningful reasoning steps. In high-dimensional continuous spaces, the model may instead rely on shortcut features or spurious representations that are unrelated to valid reasoning (Zhang et al., 2025a). This setting corresponds to **outcome supervision (OS)**, where latent states are shaped exclusively by gradients propagated from the final answer. To mitigate this limitation, existing work increasingly incorporates explicit rationales $C = (c_1, \ldots, c_N)$ as additional supervision signals. We collectively refer to these approaches as **process supervision (PS)**. Let the rationale be partitioned into $T$ reasoning steps, denoted as $S_1, \ldots, S_T$. We observe that existing process supervision methods can be understood through two complementary dimensions: *trajectory supervision* and *space supervision*.

**Trajectory Supervision.** Trajectory supervision regulates the *temporal dynamics* of latent reasoning. By requiring the accumulated latent states $L_{\leq t}$ to predict future reasoning content, such as the next explicit step $S_{t+1}$ or the future suffix $S_{>t} = (S_{t+1}, \ldots, S_T)$, trajectory supervision enforces

the sequential predictability of the latent trajectory. It therefore encourages the continuous flow to follow the logical progression of the explicit chain and provides dense signals for mitigating temporal information decay.

**Space Supervision.** Space supervision regulates the *semantic grounding* of individual latent states. It encourages each latent state $L_t$ to preserve information about its corresponding explicit step $S_t$. We distinguish two strategies based on the *space* where the alignment constraint is applied: 1) *Generative Reconstruction (GR):* This approach performs alignment in the symbolic space. It employs an auxiliary decoder to recover the corresponding textual reasoning step $S_t$ from $L_t$, thereby encouraging the latent state to retain recoverable semantic content without forcing it into a fixed embedding geometry. 2) *Geometric Compression (GC):* Conversely, this approach performs alignment directly in the latent space. Instead of generating raw tokens, it constrains $L_t$ to minimize the geometric distance to an encoded representation of $S_t$, typically using a frozen encoder as a stable semantic anchor.

## 3. The Optimization Barrier of Outcome Supervision for Latent CoT

Prior work suggests that given sufficient data scale, Transformers can implicitly internalize multi-hop reasoning structures from input-output pairs alone (Yao et al., 2025). This observation naturally extends to the design of Latent CoT: since continuous hidden states effectively provide additional computational depth, it is intuitively plausible that minimizing outcome loss alone should automatically pressure the model to utilize this capacity for reasoning.

Based on this intuition, we formulate the baseline hypothesis for latent CoT induction: *Given a sufficiently capable backbone and large-scale data, task-relevant latent dynamics should emerge solely from outcome supervision.* To rigorously test this, following prior work (Hao et al., 2025), we employ a GPT-2 (Radford et al., 2019) backbone trained on the augmented GSM8k (Cobbe et al., 2021) dataset (GSM8k-Aug, Deng et al. 2023). We minimize the standard negative log-likelihood over the final answer $y$:

$$\mathcal{L}_{\text{OS}} = -\frac{1}{|y|} \sum_{i=1}^{|y|} \log p_\theta(y_i \mid x, L, y_{<i}), \qquad (8)$$

where the continuous hidden states are implicitly integrated into the forward pass of the backbone (See Appendix A.1 and A.2 for detailed configurations).

**The Collapse of Outcome-Supervised Latent CoT.** We compare the training trajectories of Explicit CoT (Upper Bound) against a suite of Outcome-Supervised baselines.

As shown in Figure 2a and Table 1, the Explicit CoT steadily converges to viable performance level, indicating that the backbone has sufficient parameter capacity to solve these multi-step reasoning tasks when provided with explicit intermediate steps. In contrast, removing this intermediate supervision leads to a substantial performance drop [1]. Both OS-LATENT and OS-LATENT-PROJ *fail to outperform* the trivial OS-NO-COT baseline. To understand why it fails to improve performance, we analyze the underlying optimization dynamics that govern the evolution of the continuous hidden states:

**Gradient Attenuation.** To diagnose the mechanistic cause of this collapse, we analyze the gradient flow dynamics governing the latent chain. We quantify the supervision magnitude reaching each step $t$ via the gradient norm $\mathcal{G}(t)$:

$$\mathcal{G}(t) = \left\| \frac{\partial \mathcal{L}_{\text{OS}}}{\partial L_t} \right\|_2 \qquad (9)$$

Figure 2b reveals a position-wise gradient attenuation. The supervision signal is concentrated on $L_1$, while $L_{2\ldots6}$ suffers from systematic signal loss, effectively flattening into inactivity. The resulting dynamics show an early-improvement-then-degradation pattern. The validation loss decreases primarily during the initial phase when early latent states receive stronger gradients. As these gradients decay, the loss stops improving and eventually rebounds, indicating that later latent states do not receive sufficient supervision to assume the reasoning burden. This behavior is consistent with a structural shortcut: instead of distributing computation across the latent trajectory, the model relies disproportionately on early latent positions to support the final prediction. This resembles gradient starvation (Pezeshki et al., 2021), where dominant shallow features suppress the learning of deeper dependencies.

**Manifold Drift.** This structural bypass leads to an unstable evolution of the latent space. Figure 2c visualizes the PCA trajectory of the OS-Latent hidden states across training epochs, referenced against the static semantic space of Explicit CoT embeddings. The continuous hidden states exhibit significant manifold drift: since the intermediate latent states are only indirectly constrained by the final-answer loss, they lose their semantic tethering. Instead of moving toward stable semantic anchors, the trajectory diverges into an unstructured region. This confirms a fundamental geometric disconnect: without active participation in the gradient flow to constrain the search space, the optimization process fails to keep the intermediate representations close to the semantic reference region induced by explicit CoT embeddings.

---

[1] We include additional generalization experiments in Appendix C.

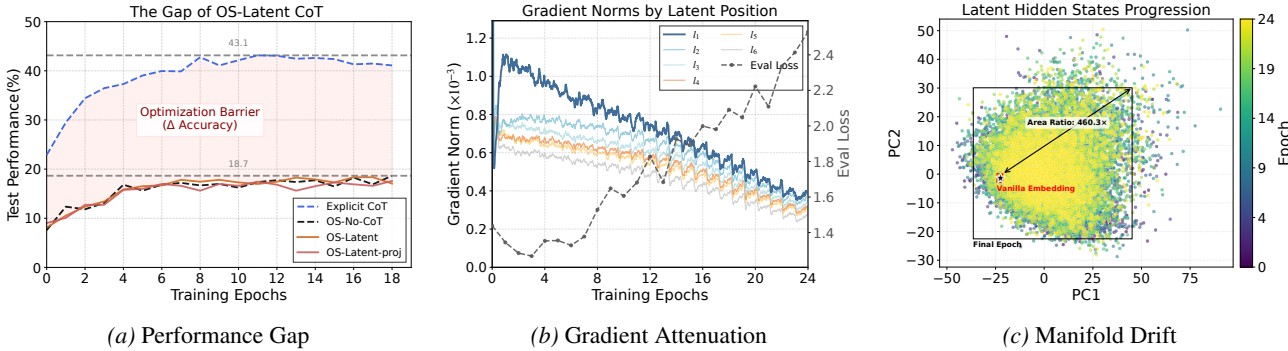

*(a)* Performance Gap        *(b)* Gradient Attenuation        *(c)* Manifold Drift

*Figure 2.* Optimization barrier of outcome supervision. (a) **Performance Gap:** Despite sufficient data, all OS variants fail to bridge the gap to the explicit CoT. (**b**) **Gradient Analysis:** The answer-level supervision signal is unevenly distributed across latent positions, with later latent states receiving weaker gradients. (**c**) **Manifold Drift:** PCA visualization of OS-Latent average last-layer hidden states across training epochs.

This analysis explains the Optimization Barrier: gradient short-circuiting precipitates manifold drift, showing that outcome supervision provides only a weak and distal surrogate for increasing the answer-level mutual information $I(L; y)$, without ensuring that individual latent states preserve valid reasoning information.

## 4. Process Supervision: Transcending the Optimization Barrier

To surmount the optimization barrier identified in Section 3, prior work (Hao et al., 2025) introduced process supervision of latent CoT, a paradigm that fundamentally alters the learning dynamics by explicitly leveraging the ground truth rationale as a structural scaffold.

### 4.1. The Scaffolding Effect: How Trajectory Supervision Reshapes Training Dynamics

At stage $k$ of total steps $T$, the model learns to generate a hybrid trajectory consisting of the first $k$ reasoning steps projected into continuous space, i.e., $L_{\leq k}$, which comprises $g \cdot k$ continuous vectors, followed by the remaining $T - k$ discrete reasoning steps. The training objective at stage $k$ is formulated as:

$$\mathcal{L}_{\text{stage-}k} = - \sum_{j=k+1}^{T} \log P_\theta \left( S_j \mid \underbrace{L_{\leq k}}_{\text{Latent CoT}}, \underbrace{S_{k+1}, \ldots, S_{j-1}}_{\text{Trajectory Scaffold}} \right),$$

$$(10)$$

where $L_{\leq k} = (L_1, \ldots, L_k)$ denotes the latent blocks generated for the first $k$ reasoning steps. (See Appendix A.2 for full training algorithms).

We leverage this progressive framework, denoted as PS-LATENT, as a controlled probe to dissect the training dynamics, and identify trajectory scaffolding and optimization shock as the coupled mechanisms driving the substantial

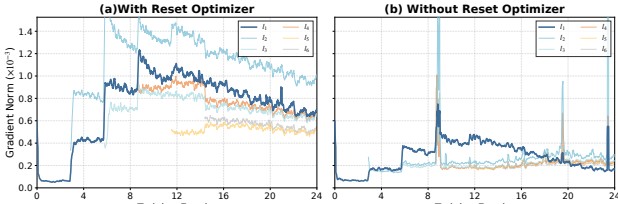

*Figure 3.* Impact of optimizer resetting on gradient dynamics. (a) With Reset: Distinct gradienzt spikes at stage transitions indicate active adaptation to new latent positions. (b) Without Reset: Stale momentum suppresses gradients.

performance recovery compared to the baseline.

Table 3 confirms that this scaffolding strategy effectively counters the optimization barrier. It allows strong supervision signals to **bridge the gradient gap inherent in deep chains**, enforcing precise state updates at the immediate generation frontier, supported by dense explicit supervision and a natural curriculum of increasing difficulty. Instead of struggling to maximize the distal mutual information with the final answer $I(L; y)$, the objective is decomposed into maximizing the local stepwise mutual information $I(L_{\leq k}; S_{k+1})$. This constraint enforces the predictability of the latent trajectory, requiring that the accumulated hidden states $L_{\leq k}$ contain sufficient information to accurately forecast the immediate reasoning step $S_{k+1}$. This injection of causal logical information significantly reduces the conditional entropy of the latent manifold, preventing it from collapsing into shortcut solutions. Beyond structural scaffolding, the *optimization state* proves critical. Since the transition to continuous vectors fundamentally shifts the loss landscape (Figure 3), retaining historical states introduces "stale momentum". Resetting the optimizer is thus essential to induce a necessary "exploration shock".

*Table 1.* Bounds & Outcome Supervision

| Method | Traj. Ctrl | Align State | Max Acc (%) |
|---|---|---|---|
| OS-NO-CoT | ✗ | – | 18.7 |
| EXPLICIT CoT | ✓ | – | 43.1 |
| OS-LATENT | ✗ | None | 9.8 / 18.3 |
| OS-GC | ✗ | MSE | 13.1$_{\downarrow 5.2}$ |
| OS-GR | ✗ | Rec | 18.2$_{\downarrow 0.1}$ |

*Table 2.* Impact of Granularity ($g$). Ablation study on information density under PS-GR.

| Granularity | $g$ | Stage 1 | | | Stage 2 | | |
|---|---|---|---|---|---|---|---|
| | | E0 | E1 | E2 | E3 | E4 | E5 |
| TOKEN-LEVEL | $\sim 1$ | 39.4$_{\uparrow 9.0}$ | 41.2$_{\uparrow 7.9}$ | 41.5$_{\uparrow 6.4}$ | 39.4$_{\uparrow 11.9}$ | 40.8$_{\uparrow 10.2}$ | 42.4$_{\uparrow 10.8}$ |
| STEP-LEVEL | $g=3$ | 33.8$_{\uparrow 3.4}$ | 36.2$_{\uparrow 2.9}$ | 36.0$_{\uparrow 0.9}$ | 32.5$_{\uparrow 5.0}$ | 34.7$_{\uparrow 4.1}$ | 36.1$_{\uparrow 4.5}$ |
| | $g=2$ | 31.7$_{\uparrow 1.3}$ | 35.0$_{\uparrow 1.7}$ | 34.8$_{\downarrow 0.3}$ | 30.7$_{\uparrow 3.2}$ | 33.5$_{\uparrow 2.9}$ | 35.1$_{\uparrow 3.5}$ |
| | $g=1$ | 30.4 | 33.3 | 35.1 | 27.5 | 30.6 | 31.6 |

*Table 3.* Dynamics of Trajectory and Space Supervision.

| Method | State Align | Setting | Stage-wise Peak Accuracy (%) | | | | | | | | Max Acc (%) |
|---|---|---|---|---|---|---|---|---|---|---|---|
| | | | S1 | S2 | S3 | S4 | S5 | S6 | S7 | S8 | |
| OS-PROGRESSIVE | None | - w/o PS | 13.0 | 16.9 | 17.2 | 18.1 | 18.7 | 18.4 | 18.3 | 18.2 | 18.7 |
| PS-LATENT | None | - w/ Reset | 25.2 | 26.1 | 26.2 | 26.8 | 28.7 | 31.2 | 30.3 | 29.6 | 31.2 |
| | | - w/o Reset | 23.7 | 21.2 | 21.8 | 22.7 | 24.0 | 24.1 | 24.7 | 24.7 | 24.7$_{\downarrow 6.5}$ |
| PS-GC | MSE | – | 20.2 | 22.0 | 22.8 | 23.1 | 23.0 | 23.2 | 23.2 | 23.0 | 23.2$_{\downarrow 8.0}$ |
| PS-GR | Rec | - $g=1$ | **35.1** | 31.6 | 30.7 | 32.4 | 33.2 | 33.9 | 34.9 | 34.5 | 34.9$_{\uparrow 3.7}$ |
| | | - $g=2$ | 35.0$_{\downarrow 0.1}$ | **35.1**$_{\uparrow 3.5}$ | **35.6**$_{\uparrow 4.9}$ | **36.5**$_{\uparrow 4.1}$ | **38.1**$_{\uparrow 4.9}$ | **38.3**$_{\uparrow 4.4}$ | **38.5**$_{\uparrow 3.6}$ | **39.1**$_{\uparrow 4.6}$ | **39.1**$_{\uparrow 7.9}$ |

## 4.2. Beyond Supervision: Calibrating Latent Reasoning via Space Supervision

PS-LATENT still trails the Explicit CoT upper bound. This signals that while trajectory supervision resolves the *optimization* bottleneck, a *representation* bottleneck persists. To address the representation bottleneck and mitigate semantic drift, prior research (Shen et al., 2025; Tan et al., 2025; Cheng & Durme, 2024) has investigated various regularization mechanisms designed to ground evolving continuous hidden states in the established semantics of the discrete vocabulary. However, a fundamental challenge remains: unlike discrete tokens, continuous hidden states lack directly golden signals and their optimal trajectories are inherently unobservable. To systematically dissect the mechanisms used to navigate this unobservable landscape, we collectively define representative approaches as an auxiliary *Space Supervision* objective, categorized into two paradigms based on the **modality of the supervision signal**. If the constraint is applied directly to the geometry of the hidden state, we term it GEOMETRIC COMPRESSION. This approach enforces a rigid topological alignment by minimizing the distance between the generated continuous hidden states $L_t$ and the ground truth average embedding per step $e_{c_t}$:

$$\mathcal{L}_{\text{GC}}(L_t) = \|L_t - e_{S_t}\|_2^2. \tag{11}$$

Alternatively, if the grounding is achieved via an indirect reconstruction signal, we term it GENERATIVE RECONSTRUCTION. These methods employs a specialized decoder $D_\psi$ to recover the original discrete token from the continuous hidden states (Wei et al., 2025; He et al., 2025), preserving semantic information without enforcing strict geometric conformity:

$$\mathcal{L}_{\text{GR}} = -\log D_\psi(S_t \mid L_t) \tag{12}$$

Here, the decoder $D_\psi$ is fully trainable and optimized jointly with the main backbone. The gradients derived from the alignment objective backpropagate directly through the $L_t$, explicitly shaping the latent representation of the latent CoT model to satisfy these semantic constraints (See Appendix A.3 for full training algorithms).

Integrating this with the trajectory control framework (See Eq. (10)), the final joint optimization objective at stage $k$ is defined as:

$$\mathcal{L} = \mathcal{L}_{\text{stage-}k} + \lambda \sum_{t=1}^{k} \mathcal{L}_{\text{Align}}(L_t, S_t). \tag{13}$$

where $\mathcal{L}_{\text{Align}} \in \{\mathcal{L}_{\text{GC}}, \mathcal{L}_{\text{GR}}\}$ denotes the chosen alignment loss computed for each generated continuous hidden state.

**The Pitfalls of Rigid Geometric Alignment.** We analyze the effectiveness of these strategies and the performance is shown in Table 3, where PS-GC leads to distinct performance degradation. To assess target validity, we conduct an **oracle experiment** measuring the upper bound of the compression target $e_{c_t}$, where the model is trained to predict the answer conditioned solely on these "ground truth" embeddings (Appendix E); the resulting peak performance (41.23%) is inferior even to the Explicit CoT baseline which generates the full reasoning chain, confirming that compressing a reasoning path into a fixed geometric point inherently incurs severe information loss. Regarding generation optimization, it is geometrically insufficient relying on MSE loss: in high-dimensional spaces, minimizing Euclidean distance fails to constrain directional alignment, allowing errors to be dispersed across dimensions or shifted into irrelevant subspaces despite a low element-wise loss. Consequently, we determine that rigid geometric compression

against static targets constitutes an ill-suited paradigm for space supervision, as it suffers from both *informational upper-bound limitations* and *optimization inefficacy*.

**Semantic Tethering via Generative Reconstruction.** In contrast to rigid geometric constraints, PS-GR achieves substantial improvements by employing a generative decoder that acts as a "semantic tether," preserving the semantic fidelity of the latent chain without artificially restricting its geometric manifold. This naturally raises a fundamental question: what is the optimal **information density** for these continuous hidden states? We investigate this by modulating the granularity $g$ to analyze the impact of information density, as detailed in Table 2. The results reveal that performance progressively improves as granularity increases, indicating that overly compressing reasoning steps induces a severe information bottleneck. However, a trade-off exists between computational efficiency and representational capacity. Thus, the optimal granularity must be balanced against the dataset's inherent difficulty, the magnitude of its reasoning jumps and the capability of model itself. To probe the limits of this approach, we conducted an **upper-bound experiment** at the maximum granularity where every discrete token is substituted with a continuous state. We find that while performance converges towards the Explicit CoT baseline, it fails to surpass it. This suggests that structural imitation is inherently bounded by the quality of the explicit supervision, implying that to transcend this limit, the model must fully internalize the latent CoT paradigm, potentially unlocking capabilities that go beyond mere reproduction in future training stages. We also provide an analysis of Space Supervision performance and underlying mechanisms under OS settings in Appendix B.

From an information-theoretic perspective, the fundamental objective of Space Supervision is to maximize the reconstructibility of the latent state, formally defined as maximizing the mutual information $I(L_t; S_t)$ between the continuous representation and the explicit rationale. In this context, GC proves theoretically deficient; minimizing Euclidean distance serves as a rigid, low-fidelity proxy that fails to guarantee information retention in high-dimensional manifolds, explaining the observed performance degradation. Conversely, GR is theoretically rigorous: its objective effectively minimizes the conditional entropy $H(S_t|L_t)$, thereby maximizing the variational lower bound of $I(L_t; S_t)$.

# 5. Deciphering Latent CoT Reasoning: An Information-Theoretic View

While Section 4 establishes the efficacy of trajectory supervision and space supervision, a critical uncertainty remains regarding the internal mechanics: *how do varying training paradigms and constraints shape the semantic content of*

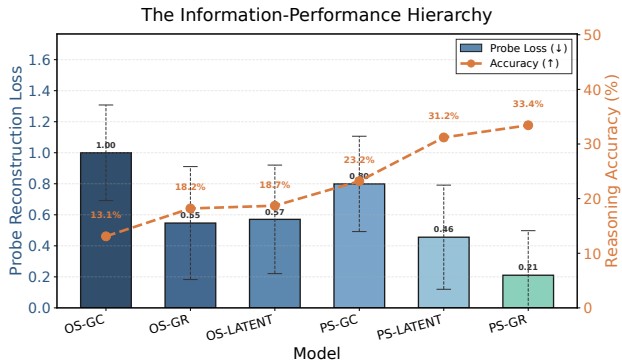

*Figure 4.* Information-performance hierarchy with ULP loss and relevant performance across different paradigms.

*these continuous hidden states?* Zhang et al. (2025a) caution that without explicit grounding, latent tokens may function merely as uninterpretable placeholders or robust shortcuts rather than faithful encoders of causal reasoning. To resolve this ambiguity and rigorously characterize the nature of the encoded information, we move *beyond surface-level performance* metrics and adopt an information-theoretic lens.

**Variational Information Bounds for Latent CoT** While the ultimate theoretical objective is to maximize mutual information with the final answer, direct estimation is intractable. Consequently, we pivot to a step-wise proxy. Let $S_t$ denote the ground-truth explicit reasoning step at time $t$. Crucially, we posit that $S_t$ somehow aligns with the proper semantic direction of the reasoning process, serving as the immutable reference for the latent state $L_t$. We quantify the quality of the latent representation by estimating the mutual information $I(L_t; S_t)$:

$$I(L_t; S_t) = H(S_t) - H(S_t \mid L_t) \qquad (14)$$

Since the marginal entropy of the ground truth $H(S_t)$ is constant for a fixed dataset, maximizing $I(L_t; S_t)$ is equivalent to minimizing the conditional entropy $H(S_t \mid L_t)$.

To address the intractability of computing $H(S_t \mid L_t)$ for continuous variables, we introduce a *variational upper bound* using a parametric probe $q_\phi(S_t \mid L_t)$. This probe acts as a lightweight decoder approximating the true posterior distribution. Its expected negative log-likelihood serves as a rigorous proxy for the conditional entropy:

$$\mathcal{L}_{\text{Info}}(L_t, S_t) = \mathbb{E}_{L_t \sim \pi}[-\log q_\phi(S_t \mid L_t)] \geq H(S_t \mid L_t)$$

where $\pi$ represents the frozen policy of the model being evaluated. We freeze the best-performance checkpoints of all baselines and train a single *Unified Latent Probe (ULP)* architecture on their generated latent states (See Appendix D for detailed training setting and curves.). The converged reconstruction loss $\mathcal{L}_{\text{Info}}$ thus provides a quantitative metric for

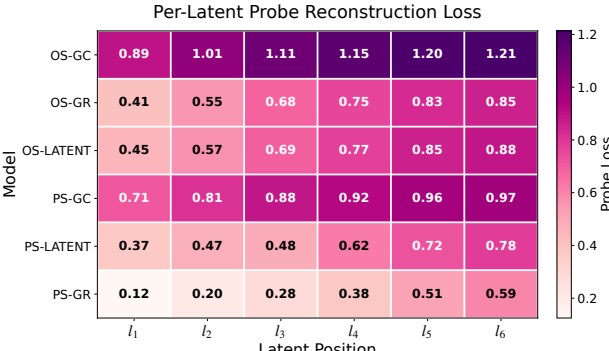

*Figure 5.* Spatiotemporal information decay, with visualization of step-wise ULP loss across the latent trajectory ($L_1 \rightarrow L_6$). A universal phenomenon of *information decay* is observed as the reasoning chain lengthens.

semantic fidelity: a lower $\mathcal{L}_{\text{Info}}$ corresponds to a tighter upper bound on conditional entropy, directly implying a higher lower bound on MI. From an optimization perspective, this establishes a direct correspondence between *reconstructibility* and *learnability*: latent states with low $\mathcal{L}_{\text{Info}}$ indicate that the model has successfully captured the rigorous semantic direction of the explicit logic, whereas high reconstruction error signals a divergence into high-entropy regions.

**The Information Hierarchy: Scaffolding as Mutual Information Maximization** Figure 4 visualizes the ULP loss $\mathcal{L}_{\text{Info}}$ alongside reasoning performance. The results reveal a striking **information hierarchy**, characterized by a rigorous inverse correlation where reasoning accuracy is strictly upper-bounded by the fidelity of information retention. OS-VARIANTS exhibit consistently high reconstruction losses, indicating a fundamental absence of reasoning-critical semantics within their latent states. Since the probe cannot extract valid logic, we infer that the optimization degenerates into encoding uninterpretable shortcut features. Even for OS-GR, which explicitly adds a reconstruction objective, fails to bridge this gap. This suggests that without the *causal logical signals*, the reconstruction loss alone is insufficient to locate the correct reasoning manifold.

Moving to PS-LATENT, we observe a significant drop in probe loss even *without space supervision*. This validates the theoretical impact of trajectory supervision: by optimizing the local stepwise objective $I(L_{\leq t}; S_{t+1})$, the training process inherently constrains the latent manifold to be predictive of the immediate reasoning step. This causal pressure forces the continuous states to retain significant semantic information to satisfy the next-token prediction requirement. Building upon this foundation, PS-GR achieves the optimal frontier. By introducing the generative decoding objective, it maximizes the variational lower bound of $I(L_t; S_t)$. This ensures that the model *internalizes* the scaf-

folding, transmuting the explicit rationale into a *semantics-preserving latent format that retains full logical fidelity*. PS-GC also suffers from severe information decay because its rigid Euclidean alignment functions as a *destructive constraint*, forcing the high-dimensional reasoning manifold to collapse onto static embeddings, thereby obliterating the fine-grained semantic fidelity required for reconstruction.

To dissect the spatio dynamics of reasoning, Figure 5 tracks the evolution of ULP loss across the latent trajectory. We observe a universal phenomenon of *information decay*, where reconstruction error progressively accumulates as the chain lengthens. This reflects an inherent vulnerability of autoregressive latent generation: unlike training under teacher forcing, where ground-truth signals correct deviations at every step, inference allows minor errors to cascade, driving the trajectory into high-entropy regions. This decay is catastrophic for unaligned or geometrically constrained baselines, which suffer from rapid semantic collapse as their states drift effectively detached from the logical path. In this context, the generative reconstruction objective functions as a crucial recurrent calibration mechanism. By enforcing reconstructibility at each transition, it effectively "resets" the accumulating semantic drift, distilling noisy hidden states back into clear, calibrated units that ensure the entire trajectory remains a faithful carrier of the reasoning process.

## 6. Related Work

**From Explicit to Latent Chain-of-Thought.** Chain-of-Thought has significantly advanced the reasoning capabilities by enabling the decomposition of complex problems into explicit intermediate steps (Wei et al., 2022). Early research (Goyal et al., 2024; Wang et al., 2024; Chen et al., 2025c), demonstrates that allocating intermediate computation enhances reasoning capacity even without meaningful semantics (Pfau et al., 2024; Shi et al., 2025). Building on this, the field has shifted towards Latent Chain-of-Thought, which fully internalizes reasoning into hidden states (Hao et al., 2025; Shen et al., 2025), with variants exploring parallelization (Wu et al., 2025), test-time scaling (You et al., 2026) and kv-cache distillation (Kuzina et al., 2025). Complementing these horizontal structures, recursive and looped architectures have also emerged to scale reasoning depth through iterative state (Geiping et al., 2025; Wang et al., 2025; Jolicoeur-Martineau, 2025; Zhu et al., 2025b). A central challenge in Latent CoT is how to supervise opaque continuous states. Existing approaches can be broadly viewed as forms of space supervision. Geometric compression (GC) constrains latent states to match target representations, such as average embeddings of reasoning steps (Cheng & Durme, 2024; Tan et al., 2025). Generative reconstruction (GR), in contrast, uses an auxiliary decoding objective to preserve the recoverability of explicit reasoning steps

from latent states (Wei et al., 2025; He et al., 2025). This distinction connects Latent CoT to broader representation learning objectives. GC is analogous to JEPA-style representation prediction, where models are trained to predict target representations in latent space rather than reconstruct raw observations (LeCun & Courant, 2022). GR is closer in spirit to reconstruction-based objectives such as masked autoencoders, which recover the original input from a learned representation (He et al., 2022).

**Internal Dynamics and Theoretical Perspectives.** Ton et al. (2025) analyses characterize Explicit CoT as a potentially inefficient channel, where generated rationales often suffer from significant redundancy and low information gain. Transcending the information bottleneck of discrete symbols, continuous representations offer provable advantages in efficiency: Zou et al. (2025) demonstrate that they achieve exponentially higher information transmission rates than discrete tokens. To address the resulting "black box" opacity, theoretical studies postulate that high-dimensional latent spaces enable *reasoning by superposition* (Zhu et al., 2025a; Xu & Sato, 2025). This mechanism allows models to process parallel logic paths simultaneously, a capability fundamentally unattainable by sequential discrete decoding (Zhang et al., 2025b). However, empirical scrutiny challenges this theoretical optimism: recent causal and adversarial analyses reveal that without robust grounding, latent tokens frequently degenerate into uninterpretable shortcuts or heuristic pattern-matching rather than encoding faithful step-by-step logic (Lin et al., 2025; Zhang et al., 2025a; Yu, 2025), highlighting the critical difficulty in verifying the causal validity of internalized thought.

## 7. Conclusion

Our analysis establishes structural scaffolding as the prerequisite for effective supervision, modeled here as mutual information maximization. In trajectory control, we find that outcome supervision degenerates into shortcuts due to unconstrained optimization, whereas Process Supervision succeeds by maximizing local stepwise information to minimize conditional entropy, effectively retaining predictability within the latent manifold. Regarding space supervision, we expose that rigid Geometric Compression acts as a destructive constraint, collapsing the high-dimensional reasoning manifold onto sparse static points. In contrast, Generative Reconstruction serves as a flexible semantic tether; by optimizing reconstructibility, it preserves intrinsic dimensionality. Ultimately, we confirm a rigorous information-performance binding: reasoning capability is strictly bounded by the mutual information retained in the latent chain. While autoregressive generation inherently suffers from spatiotemporal information decay, effective scaffolding periodically "resets" semantic drift, ensuring the trajectory remains a high-fidelity carrier of logical process.

**Limitations.** Our current analysis is bounded by the model scale (GPT-2) and task specificity, necessitating further validation on larger foundation models and broader domains. Additionally, the reliance on process annotations restricts immediate scalability, while our MI estimation remains inherently constrained by the variational probe's capacity, potentially providing a conservative lower bound on the true information content.

## Acknowledgement

We thank EIT and IDT High Performance Computing Center for providing computational resources for this project. This work was supported by the General Research Fund (GRF) of the Research Grants Council of Hong Kong (PolyU 15209724), and the 2035 Key Research and Development Program of Ningbo City under Grant No.2024Z123 and No. 2025Z034.

## Impact Statement

This work advances the field of efficient and interpretable Large Language Models by establishing a theoretical framework for Latent Chain-of-Thought reasoning. Our findings have two primary societal implications. First, regarding Computational Efficiency, our validation of high-fidelity latent reasoning supports the development of models that decouple reasoning capability from generation length. This offers a pathway toward "Green AI," significantly reducing the energy consumption and latency associated with verbose Explicit CoT without sacrificing performance. Second, regarding Safety and Interpretability, a significant concern with latent reasoning is its potential to become an unverifiable "black box." Our work mitigates this risk by demonstrating that decodability is a functional prerequisite for performance. By promoting architectures grounded in mutual information maximization, we ensure that latent thought processes remain auditable and aligned with human logic, thereby discouraging the emergence of opaque or deceptive reasoning patterns in future autonomous systems.

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

# Appendix

## A. Experimental Details

### A.1. Datasets and Hyperparameters for Main Experiments

**Dataset.** We conduct all experiments on an augmented version of the GSM8K-Aug dataset (Deng et al., 2023). The training set contains 385,620 samples, while validation and test sets retain their original sizes. Table 4 (Left) provides a formatting example.

**Base Model & Training.** All models are initialized from a GPT-2 (124M) checkpoint pre-trained on full chain-of-thought sequences for 11 epochs. We use the AdamW optimizer with a learning rate of $10^{-4}$ and an effective batch size of 128. For **progressive methods**, the optimizer state is reset at each stage transition.

*Table 4.* **Dataset Example and Common Hyperparameters.**

| Field | Content |
|---|---|
| Q | Out of 600 employees in a company, 30% got promoted while 10% received bonus. How many employees did not get either a promotion or a bonus? |
| Steps | ["<<600*30/100=180>>", "<<600*10/100=60>>", "<<180+60=240>>", "<<600-240=360>>"] |
| Ans | 360 |

| Hyperparameter | Value |
|---|---|
| Base Model | GPT-2 (124M) |
| Optimizer | AdamW |
| Learning Rate | $1 \times 10^{-4}$ |
| Weight Decay | 0.01 |
| Batch Size | 32 (Eff: 128) |
| Max Epochs | 25 |
| Precision | FP32 |
| *Progressive Settings* | |
| Epochs per Stage | 3 |
| Max Latent Tokens | 6 |

**Model-Specific Configurations.** Table 5 details the unique settings for each paradigm. **OS-variants** use a fixed latent structure, while **PS-variants** employ a progressive training strategy.

*Table 5.* **Model-Specific Configurations.** $g$ denotes latent tokens per step.

| Model | Schedule | Tokens ($g$) | Alignment | Loss Weight | Aux Decoder |
|---|---|---|---|---|---|
| OS-No-CoT | Fixed | – | – | – | – |
| OS-LATENT | Fixed | Fixed $N = 6$ | – | – | – |
| PS-LATENT | Progressive | $g = 1$ | – | – | – |
| PS-GC | Progressive | $g = 1$ | MSE | $\lambda_{\text{GC}} = 1.0$ | ✗ |
| PS-GR | Progressive | $g = 1$ | Recon | $\lambda_{\text{GR}} = 1.0$ | GPT-2 |
| PS-GR (Ext) | Progressive | $g = 2$ | Recon | $\lambda_{\text{GR}} = 1.0$ | GPT-2 |

**Progressive PS Schedule.** We adopt a **progressive schedule** (Hao et al., 2025) as PS strategy: training starts by compressing the first reasoning step into $g$ latent tokens and incrementally incorporates subsequent steps every 3 epochs until the full sequence is internalized, after which the model is fine-tuned with full capacity for the remaining epochs.

### A.2. Pseudocode for Outcome and Process Supervision

This section provides the pseudocode for the two primary supervision strategies described in Section 3 and 4.1.

**Outcome Supervision (OS-LATENT).** The OS strategy initializes a fixed sequence of $N$ latent tokens. The model must autonomously discover internal representations that maximize the likelihood of the final answer $a$, as there are no intermediate path constraints. The loss is computed solely on the answer tokens (Algorithm 1).

**Trajectory Supervision (PS-LATENT).** This strategy employs a Progressive Schedule (Algorithm 2). Training starts in Hybrid Mode, where the first $m$ steps are compressed into latent tokens while the remaining explicit steps are supervised

to scaffold the trajectory. As training progresses, more steps are internalized until the model reaches Latent Mode, where supervision applies only to the final answer.

---

**Algorithm 1** Outcome Supervision (OS-LATENT)

---

1: **Input:** Question $q$, Answer $a$, Total latent tokens $N$
2: **Output:** Loss $\mathcal{L}_{\text{OS}}$
3: $\mathbf{x}_q \leftarrow \text{TOKENIZE}(q)$
4: $\mathbf{x}_l \leftarrow [\texttt{<|start-latent|>}, \underbrace{\texttt{<|latent|>}, \ldots, \texttt{<|latent|>}}_{N}, \texttt{<|end-latent|>}]$
5: $\mathbf{x}_a \leftarrow \text{TOKENIZE}(\text{``### ''} + a + \texttt{<eos>})$
6: $\mathbf{x} \leftarrow [\mathbf{x}_q; \mathbf{x}_l; \mathbf{x}_a]$
7:
8: $\mathbf{y} \leftarrow [\underbrace{-100, \ldots, -100}_{|\mathbf{x}_q|+|\mathbf{x}_l|}; \mathbf{x}_a]$
9: $\hat{\mathbf{y}} \leftarrow \text{LATENTLM}(\mathbf{x})$
10: $\mathcal{L}_{\text{OS}} \leftarrow \text{CROSSENTROPY}(\hat{\mathbf{y}}, \mathbf{y})$
11: **return** $\mathcal{L}_{\text{OS}}$

---

**Algorithm 2** Trajectory Supervision with Progressive Schedule (PS-LATENT)

---

1: **Input:** Question $q$, Reasoning Steps $\mathcal{S} = \{S_1, \ldots, S_K\}$, Answer $a$
2: **Input:** Current epoch $e$, Epochs per stage $E_s$, Granularity $g$
3: **Output:** Loss $\mathcal{L}_{\text{PS}}$
4: $s \leftarrow \lfloor e/E_s \rfloor$
5: $m \leftarrow s + 1$
6: $N_{\text{latent}} \leftarrow m \times g$
7: $\mathbf{x}_q \leftarrow \text{TOKENIZE}(q)$
8: $\mathbf{x}_l \leftarrow [\texttt{<|start-latent|>}, \underbrace{\texttt{<|latent|>}, \ldots, \texttt{<|end-latent|>}}_{N_{\text{latent}}}]$
9: **if** $m < K$ **then**
10: $\quad \mathbf{x}_r \leftarrow \text{TOKENIZE}(S_{m+1}, \ldots, S_K)$
11: $\quad \mathbf{x}_a \leftarrow \text{TOKENIZE}(\text{``### ''} + a + \texttt{<eos>})$
12: $\quad \mathbf{x} \leftarrow [\mathbf{x}_q; \mathbf{x}_l; \mathbf{x}_r; \mathbf{x}_a]$
13: $\quad \mathbf{y} \leftarrow [\underbrace{-100, \ldots}_{|\mathbf{x}_q|+|\mathbf{x}_l|}; \mathbf{x}_r; \mathbf{x}_a]$
14: **else**
15: $\quad \mathbf{x}_a \leftarrow \text{TOKENIZE}(\text{``### ''} + a + \texttt{<eos>})$
16: $\quad \mathbf{x} \leftarrow [\mathbf{x}_q; \mathbf{x}_l; \mathbf{x}_a]$
17: $\quad \mathbf{y} \leftarrow [\underbrace{-100, \ldots}_{|\mathbf{x}_q|+|\mathbf{x}_l|}; \mathbf{x}_a]$
18: **end if**
19:
20: $\hat{\mathbf{y}} \leftarrow \text{LATENTLM}(\mathbf{x})$
21: $\mathcal{L}_{\text{PS}} \leftarrow \text{CROSSENTROPY}(\hat{\mathbf{y}}, \mathbf{y})$
22: **return** $\mathcal{L}_{\text{PS}}$

---

### A.3. Pseudocode for Space Supervision Mechanisms

This section details the two geometric constraint mechanisms introduced in Section 4.2: **Geometric Compression (GC)** and **Generative Reconstruction (GR)**. Both operate on top of the PS framework (Algorithm 2), adding an auxiliary loss to shape the latent representation space.

The **GC** mechanism enforces a geometric prior by aligning the *representative* latent state (typically the last token of the block) to the centroid of the corresponding step token embeddings via MSE loss. This encourages the latent space to mirror the geometry of the explicit reasoning space (Algorithm 3).

The **GR** mechanism employs an auxiliary decoder to reconstruct the explicit reasoning tokens from latent hidden states. This enforces that the latent representation preserves sufficient semantic content to regenerate the original step (Algorithm 4).

---

**Algorithm 3** Geometric Compression (PS-GC)

---

1: **Input:** Latent hidden states $\mathbf{H}_l = \{\mathbf{h}_1, \ldots, \mathbf{h}_{N_{\text{total}}}\}$
2: **Input:** Steps to compress $\mathcal{S}_{\text{comp}} = \{S_1, \ldots, S_m\}$, Embedding table $\mathbf{E}$, Granularity $g$
3: **Output:** Alignment loss $\mathcal{L}_{\text{GC}}$
4: $\mathcal{L}_{\text{GC}} \leftarrow 0$
5:
6: **for** $i = 1$ to $m$ **do**
7: $\quad \mathbf{t}_i \leftarrow \text{TOKENIZE}(S_i)$
8: $\quad \bar{\mathbf{e}}_i \leftarrow \frac{1}{|\mathbf{t}_i|} \sum_j \mathbf{E}[\mathbf{t}_i^{(j)}]$
9: $\quad k \leftarrow i \times g$
10: $\quad \mathbf{h}_{\text{rep}} \leftarrow \mathbf{H}_l[k]$
11: $\quad \mathcal{L}_{\text{GC}} \leftarrow \mathcal{L}_{\text{GC}} + \|\mathbf{h}_{\text{rep}} - \bar{\mathbf{e}}_i\|^2$
12: **end for**
13:
14: $\mathcal{L}_{\text{GC}} \leftarrow \mathcal{L}_{\text{GC}}/m$
15: $\mathcal{L}_{\text{total}} \leftarrow \mathcal{L}_{\text{PS}} + \lambda_{\text{GC}} \cdot \mathcal{L}_{\text{GC}}$
16: **return** $\mathcal{L}_{\text{total}}$

---

## B. Why Space Supervision Fails under Outcome Supervision?

In Section 4.2, we demonstrated that GR significantly enhances performance when paired with PS, serving as a semantic tether. However, a natural question arises: *Can this semantic tethering mechanism independently rescue OS from its optimization barrier?*

Intuitively, one might hypothesize that the auxiliary reconstruction objective $\mathcal{L}_{\text{GR}}$ would inject rich semantic gradients into every latent state, thereby counteracting the "Gradient Attenuation" and "Primacy Bias" observed in the vanilla OS-LATENT baseline.

**Empirical Failure.** Contrary to this intuition, experimental results (Table 1) reveal that Space Supervision strategies fail to yield any meaningful improvement under the OS paradigm:OS-GC harms performance, confirming that imposing rigid geometric constraints on an unguided trajectory merely distorts the manifold. Despite adding a decoder to enforce semantic content, the performance of OS-GR is indistinguishable from the trivial OS-NO-COT baseline and the vanilla OS-LATENT.

**Mechanistic Analysis: The Persistence of Gradient Decay.** To understand why the auxiliary reconstruction signal fails to correct the learning trajectory, we visualize the gradient norm distribution across latent positions for OS-GR versus the vanilla OS-LATENT baseline in Figure 6.

The visualization reveals a critical pathology: even with the auxiliary reconstruction loss, OS-GR (Left) exhibits a gradient distribution strikingly similar to the unaligned OS-LATENT (Right).

- **Primacy Bias Persists:** In both settings, the first latent token $L_1$ dominates the gradient flow, while subsequent tokens $(L_2 \ldots L_6)$ suffer from decay.

- **Decoupled Optimization:** Although $\mathcal{L}_{\text{GR}}$ provides direct supervision to each $L_t$, these gradients function as *local* alignment signals rather than *global* reasoning signals. The model learns to encode just enough information in $L_t$ to satisfy the local reconstruction decoder to minimize $\mathcal{L}_{\text{GR}}$, but the main backbone continues to bypass these deep states for the final answer prediction $y$.

This analysis highlights a fundamental theoretical distinction: **Space Supervision ensures semantic recoverability, but it does not guarantee causal utilization.** Without the Trajectory Scaffolding provided by Process Supervision to enforce the *predictive* dependency $L_t \rightarrow L_{t+1}$, the model treats the latent states as isolated "memory banks" for reconstruction rather than steps in a coherent reasoning chain. Thus, structural scaffolding is the prerequisite for effective alignment.

---

**Algorithm 4** Generative Reconstruction (PS-GR)

---

1: **Input:** Latent hidden states $\mathbf{H}_l = \{\mathbf{h}_1, \ldots, \mathbf{h}_{N_{\text{total}}}\}$
2: **Input:** Steps to reconstruct $\mathcal{S}_{\text{comp}} = \{S_1, \ldots, S_m\}$, Granularity $g$
3: **Output:** Reconstruction loss $\mathcal{L}_{\text{GR}}$
4: $\mathcal{L}_{\text{GR}} \leftarrow 0$
5: $n_{\text{valid}} \leftarrow 0$
6:
7: **for** $i = 1$ to $m$ **do**
8:     $\mathbf{t}_i \leftarrow \text{TOKENIZE}(S_i)$
9:     *// Construct latent input block*
10:
11:     **if** $g = 1$ **then**
12:         $\mathbf{z}_i \leftarrow \mathbf{H}_l[i]$
13:     **else**
14:         $\mathbf{z}_i \leftarrow [\mathbf{H}_l[(i-1)g+1]; \ldots; \mathbf{H}_l[i \cdot g]]$
15:     **end if**
16:
17:     $\text{Input} \leftarrow [\mathbf{z}_i; \mathbf{E}[\mathbf{t}_i^{(1)}]; \ldots; \mathbf{E}[\mathbf{t}_i^{(|\mathbf{t}_i|-1)}]]$
18:     $\hat{\mathbf{y}}_i \leftarrow \text{AUXDECODER}(\text{Input})$
19:     $\mathcal{L}_{\text{GR}} \leftarrow \mathcal{L}_{\text{GR}} + \text{CROSSENTROPY}(\hat{\mathbf{y}}_i, \mathbf{t}_i, \text{reduction='sum'})$
20:     $n_{\text{valid}} \leftarrow n_{\text{valid}} + |\mathbf{t}_i|$
21: **end for**
22:
23: $\mathcal{L}_{\text{GR}} \leftarrow \mathcal{L}_{\text{GR}}/n_{\text{valid}}$
24: $\mathcal{L}_{\text{total}} \leftarrow \mathcal{L}_{\text{PS}} + \lambda_{\text{GR}} \cdot \mathcal{L}_{\text{GR}}$
25: **return** $\mathcal{L}_{\text{total}}$

---

## C. Additional Generalization Experiments for Outcome Supervision

We provide additional experiments to examine whether the optimization barrier of outcome supervision is specific to the backbone scale or the arithmetic reasoning setting used in the main experiments. We first scale the backbone from GPT-2 to LLaMA-3.2-1B-Instruct on GSM8K-Aug. We then evaluate the same supervision paradigms on ProntoQA (Saparov & He, 2023), a propositional logic QA benchmark requiring multi-hop deductive reasoning. Together, these results show that the failure pattern of outcome-supervised Latent CoT is not confined to a small backbone or to arithmetic reasoning.

### C.1. Scale-Up Experiment on LLaMA-3.2-1B-Instruct

To examine whether the optimization barrier of outcome supervision is specific to the small GPT-2 backbone used in our main experiments, we conduct an additional scale-up experiment with LLaMA-3.2-1B-Instruct on GSM8K-Aug. This setting uses a substantially larger and more modern backbone while keeping the same comparison between explicit CoT, outcome-supervised Latent CoT, and answer-only training.

We train all models with the AdamW optimizer using a learning rate of $1\mathrm{e}{-4}$, weight decay $0.01$, a cosine learning-rate schedule with warmup, gradient clipping at $1.0$, effective batch size $128$, and BF16 precision. Experiments are run on 4 RTX 5880 Ada GPUs.

*Table 6.* Scale-up experiment on LLaMA-3.2-1B-Instruct with GSM8K-Aug. Outcome-supervised Latent CoT does not materially outperform the answer-only baseline, while explicit CoT remains substantially stronger.

| Method | Best Acc. | Best Epoch |
|---|---|---|
| Explicit CoT | 60.20% | 10 |
| OS-LATENT | 33.81% | 9 |
| OS-NO-COT | 32.98% | 8 |

As shown in Table 6, OS-LATENT achieves $33.81\%$ best accuracy, which is close to the answer-only baseline at $32.98\%$ and far below explicit CoT at $60.20\%$. This result reproduces the same qualitative pattern observed in the main GPT-2 experiments: adding latent states under outcome supervision alone does not reliably induce useful latent reasoning dynamics. The substantial gap between explicit CoT and OS-LATENT further indicates that the failure is not simply due to insufficient

*Figure 6.* **Gradient Norm Dynamics Comparison.** OS-GR exhibits a lower gradient magnitude than OS-LATENT. This indicates *destructive interference* between the conflicting gradients of the local reconstruction objective and the global outcome shortcut, further proving the incompatibility of GR with pure Outcome Supervision.

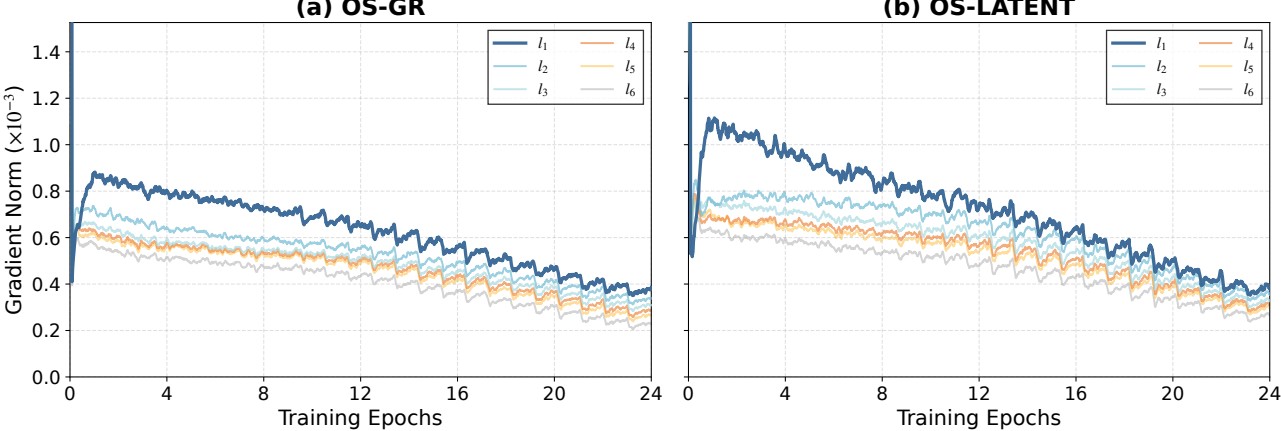

backbone capacity, since the same model benefits strongly from explicit intermediate reasoning supervision.

### C.2. Cross-Task Experiment on ProntoQA

To examine whether the observed outcome-supervision failure is specific to arithmetic reasoning, we further evaluate on ProntoQA, a propositional logic QA benchmark requiring multi-hop deductive reasoning. ProntoQA is structurally different from GSM8K-Aug: instead of numerical calculation, it requires the model to apply symbolic rules over a chain of logical implications. This makes it a useful complementary testbed for evaluating whether latent CoT supervision patterns generalize beyond arithmetic.

*Table 7.* Cross-task evaluation on ProntoQA and GSM8K-Aug. On ProntoQA, OS-LATENT remains close to the answer-only baseline, while PS-LATENT substantially improves performance. GSM8K-Aug results are included for comparison with the main experiments.

| Method | ProntoQA Acc. | GSM8K-Aug Acc. |
| --- | --- | --- |
| PS-LATENT | 97.40% | 31.2% |
| Explicit CoT | 80.80% | 43.1% |
| OS-LATENT | 78.60% | 18.3% |
| OS-NO-CoT | 78.60% | 18.7% |

As shown in Table 7, OS-LATENT obtains the same accuracy as the answer-only baseline on ProntoQA, indicating that latent states induced by outcome supervision alone do not provide a measurable benefit over direct answer prediction in this setting. In contrast, PS-LATENT reaches $97.40\%$ accuracy, substantially outperforming both OS-LATENT and explicit CoT. This result supports the same qualitative conclusion as the GSM8K-Aug experiments: outcome supervision alone is insufficient to reliably induce useful latent reasoning dynamics, whereas process supervision provides effective intermediate guidance.

We further evaluate the Unified Latent Probe (ULP) on ProntoQA. The probe loss preserves the same information hierarchy observed in the main experiments: PS-LATENT achieves lower probe loss than OS-LATENT ($0.434$ vs. $0.546$). Since lower ULP loss indicates better recovery of explicit reasoning information from latent states, this result provides additional evidence that the advantage of process supervision is reflected not only in task accuracy but also in the amount of recoverable reasoning information encoded in the latent trajectory.

# D. Training Procedure of Unified Latent-MI Probe

This appendix provides a detailed description of the Unified Latent-MI Probe (ULP) training procedure used to quantify the information content of latent representations, corresponding to the variational upper bound estimation described in Section 5.

## D.1. Hidden State Extraction

Given a trained latent CoT model with parameters $\theta$, we extract hidden states from the continuous latent chain. For a given input question $q$ and its corresponding ground-truth reasoning steps $\mathcal{S} = \{S_1, S_2, \ldots, S_K\}$, the model processes the sequence interleaved with latent blocks. For the $t$-th reasoning step $S_t$, the input context is:

$$x_t = [\ldots, S_{t-1}, \texttt{<|start-latent|>}, \underbrace{L_{t,1}, \ldots, L_{t,N}}_{N \text{ latent tokens}}, \texttt{<|end-latent|>}] \tag{15}$$

where $N = 6$ is the default number of latent tokens per step.

During the forward pass, we extract the hidden state $\mathbf{h}_t \in \mathbb{R}^d$ from the **last latent token** $L_{t,N}$ (where $d = 768$ for GPT-2 base). This vector $\mathbf{h}_t$ serves as the numerical realization of the latent state $L_t$ discussed in the main text. The extracted pairs $\{(\mathbf{h}_t, S_t)\}$ from all baselines are stored in half-precision format to construct the unified probing dataset.

## D.2. Probe Architecture

The probe aims to approximate the posterior distribution $q_\phi(S_t \mid L_t)$. We employ a **Reconstruction Probe** consisting of:

1. **Projection Layer**: A linear transformation $f_{\text{proj}} : \mathbb{R}^d \to \mathbb{R}^d$ that maps the model's latent space to the probe's embedding space:

$$\mathbf{z}_t = \mathbf{W}_{\text{proj}}\mathbf{h}_t + \mathbf{b}_{\text{proj}} \tag{16}$$

   where $\mathbf{W}_{\text{proj}}$ is initialized from $\mathcal{N}(0, 0.02^2)$ and $\mathbf{b}_{\text{proj}} = \mathbf{0}$.

2. **Conditional Decoder**: A pre-trained GPT-2 language model that generates the explicit tokens of step $S_t$ autoregressively. It utilizes a shared embedding table $\mathbf{E} \in \mathbb{R}^{|V| \times d}$.

## D.3. Training Procedure

To condition the generation on the latent state, we employ a teacher-forcing strategy. The projected latent vector $\mathbf{z}_t$ is prepended to the embeddings of the target reasoning step $S_t = [w_1, w_2, \ldots, w_M]$. The input embeddings passed to the probe decoder are:

$$\text{Input} = [\mathbf{z}_t; \mathbf{E}[w_1]; \mathbf{E}[w_2]; \ldots; \mathbf{E}[w_{M-1}]] \tag{17}$$

This architecture forces the decoder to rely on $\mathbf{z}_t$ for semantic guidance. We first define the per-sample reconstruction loss:

$$\ell(\mathbf{h}_t, S_t) = -\frac{1}{|S_t|} \sum_{j=1}^{|S_t|} \log P_\phi(w_j \mid \mathbf{z}_t, w_{<j}) \tag{18}$$

The probe parameters $\phi$ are then optimized to minimize the expected loss over the unified dataset $\mathcal{D}$:

$$\mathcal{L}_{\text{Info}} = \frac{1}{|\mathcal{D}|} \sum_{(\mathbf{h}_t, S_t) \in \mathcal{D}} \ell(\mathbf{h}_t, S_t) \tag{19}$$

Data from all latent models (OS, PS, GR, GC) are combined into a unified dataset to ensure a fair comparison. Figure 7 illustrates the training dynamics of the ULP. We observe that PS-GR achieves the fastest convergence and lowest final reconstruction loss, confirming its superior semantic fidelity. In contrast, OS-GC plateaus at a significantly higher loss, reflecting the severe information loss caused by rigid geometric constraints. The detailed training hyperparameters are listed in Table 8.

*Figure 7.* **Per-Model Probe Loss History.**

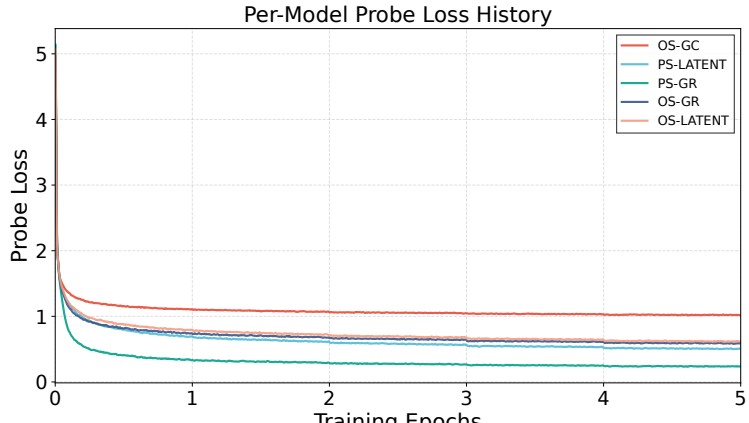

*Table 8.* **Hyperparameters for ULP Training.**

| Hyperparam | Value |
|---|---|
| Optimizer | AdamW |
| LR | $1 \times 10^{-4}$ |
| Wt. Decay | 0.01 |
| Scheduler | Linear (10%) |
| Batch Size | 128 |
| Epochs | 5 |
| Clip | 1.0 |
| Precision | FP16 |

*Table 9.* Probe reliability across architectures. All probes are trained on the same frozen hidden states. Lower values indicate better recovery of explicit reasoning information.

| Variant | 2-Layer MLP | 2L-4H Transformer |
|---|---|---|
| PS-GR | 1.310 | 0.375 |
| PS-LATENT | 1.755 | 0.722 |
| OS-GR | 1.806 | 0.775 |
| OS-LATENT | 1.861 | 0.824 |
| OS-GC | 2.145 | 1.049 |
| PS-GC | 2.194 | 1.097 |

### D.4. Probe Reliability Across Architectures

Since recovering an explicit reasoning step from a latent state is naturally a sequence-generation problem, we use a GPT-2 probe as the default decoder-based probe in the main experiments. To ensure that the observed information hierarchy is not an artifact of this particular probe architecture, we conduct an additional robustness check with three architecturally distinct probes trained on the same frozen hidden states: a two-layer MLP, and a two-layer four-head Transformer. As shown in Table 9, all three probe architectures produce the same ranking over latent representations.

## E. Oracle Embedding Experiment

To investigate whether the geometric target of GC is fundamentally capable of supporting reasoning, we design an **Oracle Embedding** experiment that provides the model with *perfect* ground-truth step embeddings at training time. The GC mechanism trains latent states to approximate the average embedding of corresponding reasoning steps. A natural question arises: *even if this alignment were perfect, would such representations be sufficient for accurate answer prediction?* Instead of learning latent representations, we directly replace each placeholder token's embedding with the oracle target—the average of ground-truth step token embeddings (Algorithm 5).

Despite providing perfect oracle embeddings (the exact target that GC attempts to approximate), the model achieves only 41.2%. This negative result reveals a fundamental limitation: *average embeddings discard critical sequential and compositional information* required for multi-step reasoning. The centroid of a step's tokens loses the ordering, syntax, and logical structure that distinguish meaningful reasoning from arbitrary token mixtures. This experiment provides strong evidence that the GC objective is inherently flawed: even perfect alignment to embedding centroids cannot recover the information necessary for reasoning.

---

**Algorithm 5** Oracle Embedding Experiment (Upper Bound for GC)

---

1: **Input:** Question $q$, Steps $\mathcal{S} = \{S_1, \ldots, S_K\}$, Answer $a$, Embedding table $\mathbf{E}$
2: **Output:** Loss $\mathcal{L}_{\text{Oracle}}$
3:
4: *// Compute ground-truth step average embeddings*
5: **for** $i = 1$ to $K$ **do**
6:     $\mathbf{t}_i \leftarrow \text{TOKENIZE}(S_i)$
7:     $\bar{\mathbf{e}}_i \leftarrow \frac{1}{|\mathbf{t}_i|} \sum_j \mathbf{E}[\mathbf{t}_i^{(j)}]$
8: **end for**
9:
10: $\mathbf{x}_q \leftarrow \text{TOKENIZE}(q)$
11: $\mathbf{x}_a \leftarrow \text{TOKENIZE}(\texttt{"\#\#\# "}\|a\|\texttt{<eos>})$
12: $\mathbf{x} \leftarrow [\mathbf{x}_q; \texttt{<placeholder>} \times K; \mathbf{x}_a]$
13:
14: $\mathbf{X}_{\text{emb}} \leftarrow \mathbf{E}[\mathbf{x}]$
15: **for** $i = 1$ to $K$ **do**
16:     $\mathbf{X}_{\text{emb}}[\text{pos}_i] \leftarrow \bar{\mathbf{e}}_i$
17: **end for**
18:
19: $\hat{\mathbf{y}} \leftarrow \text{GPT2}(\mathbf{X}_{\text{emb}})$
20: $\mathcal{L}_{\text{Oracle}} \leftarrow \text{CROSSENTROPY}(\hat{\mathbf{y}}, \mathbf{x}_a)$
21: **return** $\mathcal{L}_{\text{Oracle}}$

---

