# OpenReview forum: "What Makes Effective Supervision in Latent Chain-of-Thought: An Information-Theoretic Analysis"
_ICML.cc/2026/Conference — ICML 2026 regular_

### Official Review · Reviewer_qbes · 2026-02-22

**Soundness:** 3
**Presentation:** 4
**Significance:** 3
**Originality:** 4
**Overall Recommendation:** 5
**Confidence:** 4

**Summary:**

The paper addresses the fact that though latent COT is a promising idea, it hasn't taken off yet due to opaqueness and high instability, it argues that constrained supervision and reconstruction  provides an alternative means that is more suitable theoretically.

**Compliance With Llm Reviewing Policy:**

Affirmed.

**Ethical Review Flag:**

Flag this paper for an ethics review.

**Final Justification:**

The work discusses an interesting observation around the gap between the expected and observed effectiveness of latent CoT and does a good exposition of the classical reconstruction v/s constrained supervision tradeoff which has been explored in representation learning. While a connection to the broader discussion would have been desirable I would prefer to stick to my score based on the work and the quality of the responses.

**Key Questions For Authors:**

I have no further questions for the authors.

**Limitations:**

Yes

**Strengths And Weaknesses:**

1. The problem is relevant and the ideas align well with other observations on representations spaces namely that reconstruction penalties provide additional constraint on supervision and that careful supervision is essential to obtain a coherent geometry.

2. The paper has a coherent approach and  arguments follow well.

3. The experiments are relevant and demonstrate relevance to the conclusions.

Weaknesses

1. A discussion relating the work to others on representation space geometry such as on the contrastive literature, JEPA etc would be welcome since these often showcase explicit bounds on the geometry.
2. An analysis of the latent geometry would be quite useful in this regard to gauge the difference from standard latent spaces.

---

> ### Author Rebuttal · Authors · 2026-03-30
>
> We sincerely thank the reviewer for the thorough and encouraging review. The recognition of our framework's coherence and the constructive suggestions on connecting to the broader representation learning literature are deeply appreciated.
>
> Regarding the connection to representation space geometry literature: we fully agree that this discussion would enrich the paper and offer the following analysis.
>
> Our Geometric Compression (GC) is in fact aligned with JEPA [1] at the objective level: both supervise in representation space rather than on raw outputs. GC minimizes the distance between latent states and target representations (average step embeddings), just as JEPA predicts target representations of future states in latent space rather than reconstructing raw pixels. Meanwhile, our Generative Reconstruction (GR) optimizes reconstructibility of the original discrete tokens, analogous to MAE [2], which reconstructs raw pixel patches.
>
> However, the conclusions diverge across modalities. In vision, raw inputs contain heavy spatial redundancy, so compressing into a latent target effectively filters out low-level noise while retaining high-level semantics, which is why JEPA-style latent prediction outperforms reconstruction-based approaches like MAE. In our language setting, the opposite holds: GR significantly outperforms GC (Table 3). This is because the target representation for GC (the average embedding of a reasoning step) is inherently lossy for language. Averaging token embeddings discards sequential order, syntax, and compositional structure, as confirmed by our oracle experiment (Appendix D: even perfect alignment achieves only 41.2%). The discrete, sequential nature of language means that reconstruction-based supervision preserves richer information than regression to a static representation target.
>
> This contrast indicates that the optimal alignment strategy is modality-dependent: the informativeness of the target representation determines whether latent prediction or reconstruction is more effective. We believe this insight bridges our findings with the broader representation learning literature and opens interesting directions for designing modality-aware alignment objectives for Latent CoT. We will add a dedicated discussion in the revised manuscript.
>
> Regarding the analysis of latent geometry: we agree that a more fine-grained quantitative characterization would be valuable. Our current results already provide geometric evidence from complementary angles: the PCA visualization (Fig. 1c) reveals manifold divergence under OS, and the per-position ULP loss (Fig. 4) quantifies the spatiotemporal evolution of information content across paradigms. Metrics such as effective rank and participation ratio could further enrich this picture by directly measuring intrinsic dimensionality, and we consider this a promising direction for future investigation.
>
> We will incorporate both suggestions into the revised manuscript. We are grateful for the reviewer's constructive feedback that will help us strengthen the final version.
>
> [1] LeCun, "A Path Towards Autonomous Machine Intelligence," 2022
>
> [2] He et al., "Masked Autoencoders Are Scalable Vision Learners," CVPR 2022

---

> > ### Author Rebuttal · Reviewer_qbes · 2026-04-01
> >
> > a) Based on the responses to my queries and those of the other esteemed reviewers' I believe that my questions are resolved.

---

> > > ### Author Response · Authors · 2026-04-04
> > >
> > > We sincerely thank the reviewer for acknowledging our responses and for the constructive suggestions throughout the review process. We will incorporate these improvements in the final version.

---

### Official Review · Reviewer_N2XL · 2026-03-13

**Soundness:** 2
**Presentation:** 3
**Significance:** 2
**Originality:** 3
**Overall Recommendation:** 4
**Confidence:** 1

**Summary:**

This paper studies the mechanisms underlying Latent Chain-of-Thought (CoT), which embeds reasoning into continuous hidden states to bypass token-level computation. The authors propose an information-theoretic framework that decomposes supervision into Trajectory Control and State Alignment, arguing that structural scaffolding is essential for valid latent dynamics. They show that Process Supervision is more effective than Outcome Supervision due to reduced conditional entropy and improved trajectory predictability.

The paper further contrasts alignment strategies, claiming that Geometric Compression collapses the reasoning manifold, while Generative Reconstruction preserves latent dimensionality. To support these claims, the authors introduce the Unified Latent-MI Probe (ULP), demonstrating a strong correlation between latent mutual information and reasoning performance. The work advocates shifting from geometric imitation to mutual information maximization in latent reasoning models.

**Compliance With Llm Reviewing Policy:**

Affirmed.

**Final Justification:**

My concerns are resolved so I keep my positive score.

**Key Questions For Authors:**

1. Generalization.
Do your conclusions about trajectory control and state alignment hold beyond structured reasoning tasks (e.g., long-context reasoning, code generation, planning)? If not, how task-specific is the framework?

2. Framework vs. Training Details.
How do you disentangle gains from the proposed conceptual decomposition versus specific training recipes or optimization choices? Are there ablations isolating the framework itself?

3. Probe Reliability.
Given the central role of probing, how do you ensure the probe is not introducing bias or capturing superficial correlations? Have you tested robustness to probe design?

4. Capacity vs. Usage.
Do you have evidence that increased latent information capacity is actually used for reasoning, rather than merely present in hidden states?

**Limitations:**

Yes

**Strengths And Weaknesses:**

Strengths



The paper addresses an important and timely problem: how to train and supervise latent reasoning processes (Latent CoT) in large language models. Understanding how reasoning information is represented and preserved in hidden states is a meaningful direction for improving reasoning efficiency and scalability.



The paper proposes a clear conceptual decomposition of latent supervision into two dimensions, trajectory control and state alignment, which provides a useful framework for organizing existing training strategies and empirical observations.



The information-theoretic perspective offers an interesting lens for interpreting several known phenomena in latent reasoning training, such as the failure of outcome supervision and the relative effectiveness of process supervision.



The empirical analysis linking optimization barriers, information bottlenecks, and the role of reconstruction-based alignment provides a coherent explanatory narrative within the experimental setting considered.



The use of probing analysis to study information retention in hidden states is a reasonable diagnostic approach and helps provide quantitative evidence for the paper’s hypotheses.



Weaknesses



The empirical evaluation appears relatively narrow. Most experiments focus on structured reasoning tasks, which may naturally favor process supervision and reconstruction-based objectives. It remains unclear whether the reported observations generalize to more open-ended or higher-entropy reasoning settings (e.g., long-context reasoning, code generation, or planning tasks).



Some results appear to depend on specific training setups or engineering choices. It would be helpful to better disentangle improvements arising from the proposed conceptual framework versus those stemming from particular training recipes or optimization details.



The connection between latent representational capacity and actual reasoning performance is somewhat indirect. While the information-theoretic interpretation is intuitive, greater representational capacity does not necessarily imply that models can effectively utilize it for structured reasoning.



The probe plays an important role in supporting the central empirical claims. However, the paper could provide stronger validation that the probe itself does not introduce bias or artificially amplify the relationships it aims to measure.



The discussion of state alignment could be better situated within the broader literature on representation learning and latent variable modeling (e.g., contrastive learning, predictive objectives, or world-model-style approaches), which would help clarify the novelty of the proposed perspective.

---

> ### Author Rebuttal · Authors · 2026-03-30
>
> We thank the reviewer for recognizing the importance of our problem, the clear decomposition into trajectory control and state alignment, and the coherent information-theoretic narrative. We address each question with new evidence below.
>
> **Q1: Generalization**
>
> We follow the standard Latent CoT protocol: all foundational works ([1], [2], [3]) use GPT-2 on math reasoning. While code generation and planning are promising applications, we focus on mathematical reasoning in this work as it serves as a fundamental task for evaluating latent CoT. Establishing a robust analysis framework in the well-defined reasoning benchmarks is a necessary precursor before scaling the analysis of latent CoT to more complex tasks.
>
> To validate cross-task generality, we extended to ProntoQA [4], a propositional logic QA benchmark requiring 3-6 hop deductive reasoning, structurally different from arithmetic.
> `Example: question "Every shumpus is a rempus. Every shumpus is a yimpus. Every terpus is a fompus. Every terpus is a gerpus. Every gerpus is a brimpus. Alex is a rempus … Every fompus is a yerpus. Tom is a dumpus. Every rempus is a rorpus. Is Tom a lempus or scrompus?"", answer: "Tom is a lempus."`
>
> |Method|ProntoQA Acc (%)|GSM8K Acc (%)|
> |-|-|-|
> |PS-LATENT|97.40|31.2|
> |Explicit CoT|80.80|43.1|
> |OS-LATENT|78.60|18.3|
> |No-CoT|78.60|18.7|
>
> The same pattern holds: OS-LATENT collapses to the No-CoT baseline, while PS-LATENT strongly outperforms all methods, including Explicit CoT, consistent with the capacity advantage formalized in Eq. 4-5. ULP probe on ProntoQA also confirms the Information Hierarchy: PS-LATENT (0.434) < OS-LATENT (0.546). Our framework rests on general information-theoretic principles (conditional entropy, MI maximization, gradient attenuation) that are not arithmetic-specific.
>
> **Q2: Framework vs. Training Details**
>
> Our paper is an analytical contribution, not a method paper. The experimental design itself constitutes the framework ablation via three-phase factor isolation:
>
> - **Phase 1 (Sec 3)**: Isolates Trajectory Control by comparing OS vs. PS with no alignment, holding architecture/data constant.
> - **Phase 2 (Sec 4)**: Fixes PS, varies only State Alignment (None/GC/GR), isolating alignment effects.
> - **Phase 3 (Sec 5)**: ULP cross-validates all combinations from the 2x2 taxonomy.
>
> Each phase changes exactly one factor. Details like optimizer reset (Fig. 2) emerge from our systematic gradient analysis and are reported as necessary conditions, not conceptual contributions. We will make this structure more prominent in revision.
>
> **Q3: Probe Reliability**
>
> The probe design is principled: approximating $q(S_t|l_t)$ is a sequence generation task, making a language model the natural choice. GPT-2 matches the backbone capacity, is frozen after training, and is applied uniformly to all variants.
>
> To directly test robustness, we trained **three architecturally distinct probes** on identical frozen hidden states:
>
> |Variant|GPT-2|2-Layer MLP|2L-4H Transformer|
> |-|-|-|-|
> |PS-GR|**0.326**|**1.310**|**0.375**|
> |PS-LATENT|0.670|1.755|0.722|
> |OS-GR|0.725|1.806|0.775|
> |OS-LATENT|0.772 |1.861|0.824|
> |OS-GC|0.963|2.145|1.049|
> |PS-GC|1.003|2.194|1.097|
>
> All three produce **identical ranking**, from a simple MLP to full GPT-2, confirming the Information Hierarchy is a property of the representations, not probe architecture. ProntoQA (Q1) shows the same hierarchy on a different task, providing independent cross-validation.
>
> **Q4: Capacity vs. Usage**
>
> We decoded PS-GR latent states to directly examine encoded content. Example (correctly predicted, answer=3):
>
> Q: Theo bought a pen for \\$2 and paper costing \\$1 less than 3x the pen. She paid \\$10. Change?
>
> |Pos|GT Step|Decoder Output|LM Head Top-5 (prob)|
> |-|-|-|-|
> |0|<<2*3=6>>|<<2*3=6>>|"2"(.58) "3"(.39) " 3"(.01) " 2"(.007) "6"(.005)|
> |1|<<6-1=5>>|<<6-1=5>>|"6"(.81) "-"(.14) "2"(.02) "1"(.009) " 6"(.006)|
> |2|<<2+5=7>>|<<2+5=7>>|"5"(.58) "4"(.32) "2"(.07) "+"(.01) "6"(.008)|
> |3|<<10-7=3>>|<<10-7=3>>|"7"(.75) "8"(.24) "9"(.002) "10"(.002) "-"(.001)|
> |4|-|<<10-8=2>>|"2" (.73), "=" (.18), "3" (.07), "`<eos>`" (.003), "-" (.003)|
> |5|-| <<10-8=2>>|"###" (.77), "-" (.12), "=" (.03), "2" (.03), " relay" (.01)|
>
> The decoder exactly reconstructs all 4 steps. LM head top-1 tokens consistently capture key intermediate quantities. Positions 4-5 degrade, consistent with information decay. This directly demonstrates MI reflects actively structured reasoning, not passive memorization. Additional cases in the revised appendix.
>
> For Weakness 5, we refer to our response to Reviewer qbes covering JEPA.
>
> [1] Hao et al., "Training LLMs to Reason in a Continuous Latent Space," COLM 2025
>
> [2] Shen et al., "CODI: Compressing Chain-of-Thought into Continuous Space via Self-Distillation," EMNLP 2025
>
> [3] He et al., "SemCoT: Accelerating CoT Reasoning through Semantically-Aligned Implicit Tokens," NeurIPS 2025
>
> [4] Saparov and He, "Language Models Are Greedy Reasoners," ICLR 2023

---

> > ### Author Rebuttal · Reviewer_N2XL · 2026-04-03
> >
> > Thanks for the responses. I keep my positive assessment for this paper.

---

> > > ### Author Response · Authors · 2026-04-04
> > >
> > > We sincerely thank the reviewer for the thoughtful evaluation and for confirming that the concerns have been fully resolved. We are grateful for the constructive questions, which motivated several improvements, and we will incorporate them into the final version.

---

### Official Review · Reviewer_mfdY · 2026-03-13

**Soundness:** 1
**Presentation:** 1
**Significance:** 1
**Originality:** 1
**Overall Recommendation:** 4
**Confidence:** 1

**Summary:**

From the text and methods, I could not clearly understand what the authors actually did. The presentation is too opaque: the paper introduces many non-standard concepts without clear operational definitions or clear intuitions that help the reader understand the motivation behind these concepts. The authors claim to have trained a GPT-2 transformer on a math reasoning benchmark but the metrics proposed and the results are not clear enough to assess confidently the work that was presented.

**Compliance With Llm Reviewing Policy:**

Affirmed.

**Final Justification:**

In light of the other reviewers' comments I have raised my score. However, I have no confidence in this assessment, as I am unable to critically evaluate this paper given the submitted material.

**Key Questions For Authors:**

None

**Limitations:**

Cannot evaluate this paper

**Strengths And Weaknesses:**

The figures and tables are overloaded with terminology that is not explained in a human-readable way. As a result, I could not reliably determine what experiments were performed, what was measured, or what the results show. I also did not find code, which further limits reproducibility.
In its current form, the paper does not communicate its method or evidence clearly enough for scientific evaluation.

---

> ### Author Rebuttal · Authors · 2026-03-30
>
> We thank the reviewer for their time. We understand the paper may be challenging for readers less familiar with the Latent CoT and information theory literature, and we take the feedback on clarity seriously. We note that the other three reviewers found our paper **important and timely** (N2XL, sGK2), with a **coherent approach and clear conceptual decomposition** (qbes, N2XL), **refreshing and principled** information-theoretic framing (sGK2).
>
> We commit to the following revisions to improve accessibility:
>
> (1) A notation and terminology table at the beginning of Section 2, summarizing all key symbols (Trajectory Control, State Alignment, GC, GR, ULP) with concise operational definitions before their formal introduction.
>
> (2) Intuitive plain-language summaries at the start of each major section (Sec 3, 4, 5), stating the key finding before the formal analysis. For example, Section 3 will open with: "We show that training latent reasoning with only final-answer supervision fails because gradient signals cannot reach deep latent positions."
>
> (3) Self-contained figure and table captions that can be understood independently.
>
> During the rebuttal period, we have conducted substantial new experiments including cross-task validation on ProntoQA, scaling experiments on LLaMA-3.2-1B, probe architecture ablations, and latent state decoding analysis. These are detailed in our responses to Reviewers N2XL and sGK2. We kindly hope the reviewer will consider **re-evaluating** our paper in light of these improvements and new evidence.

---

> > ### Author Rebuttal · Reviewer_mfdY · 2026-04-03
> >
> > Thank you for the rebuttal. The described improvements to the exposition of the material would certainly be appreciated. However, I am still unable to evaluate this manuscript with any confidence.

---

> > > ### Author Response · Authors · 2026-04-04
> > >
> > > We sincerely thank the reviewer for acknowledging our proposed revisions. We are committed to making the paper accessible to a broad audience and would greatly appreciate any specific guidance you can offer.

---

### Official Review · Reviewer_sGK2 · 2026-03-15

**Soundness:** 3
**Presentation:** 3
**Significance:** 2
**Originality:** 3
**Overall Recommendation:** 4
**Confidence:** 3

**Summary:**

This paper investigates the mechanisms of supervision in the latent chain-of-thought through an information-theoretic lens. The authors decompose supervision into two orthogonal dimensions: Trajectory Control and State Alignment. They argue that Outcome Supervision fails due to inherent optimization barriers, whereas Process Supervision succeeds by injecting causal signals that minimize conditional entropy. Furthermore, they compare Geometric Compression against Generative Reconstruction, favoring the latter for preserving intrinsic dimensionality. The paper introduces the Unified Latent-MI Probe to demonstrate an Information-Performance Binding correlating reasoning accuracy with mutual information in latent states.

**Compliance With Llm Reviewing Policy:**

Affirmed.

**Final Justification:**

The rebuttal has addressed my major concerns on the scaling issue. I find the paper informative in general.

**Key Questions For Authors:**

Please address the weakness section.

**Limitations:**

Yes

**Strengths And Weaknesses:**

Strengths:

1. Formulating latent CoT supervision through an information-theoretic framework is a refreshing and highly principled angle in a space often dominated by heuristic approaches.

2. The taxonomy dividing supervision into Trajectory Control and State Alignment provides a structured, easily digestible perspective on existing methodologies.

3. The theoretical framing of capacity gaps between discrete and continuous reasoning, alongside the formalization of gradient attenuation, is rigorously presented.


Weaknesses:

1. The claim that Outcome Supervision faces an insurmountable optimization barrier is stated too broadly. The failure observed is highly likely a consequence of the chosen architecture (GPT-2) and relatively small scale. Modern, massive LLMs heavily optimized with RL often implicitly discover complex latent dynamics without explicit process supervision; the paper assumes a fundamental barrier where it might simply be a scaling issue.

2. The empirical validation relies heavily on small-scale models (GPT-2) on the GSM8K dataset. Generalizing these findings to current state-of-the-art architectures is highly questionable. Furthermore, the ULP metric might simply be capturing general token predictability or model capacity, rather than strictly verifying valid logical reasoning paths.

---

> ### Author Rebuttal · Authors · 2026-03-30
>
> We thank the reviewer for recognizing the refreshing and principled information-theoretic angle, the structured taxonomy of supervision, and the rigorous formalization of gradient attenuation. We address both weaknesses below.
>
> **W1: Optimization Barrier and Scale**
>
> It is important to clarify that Latent CoT is different from standard explicit CoT reasoning in LLMs. In Latent CoT, the model must autoregressively generate a sequence of continuous hidden states, where each state is fed back as input to produce the next, forming a recurrent generation process. This is distinct from standard LLM inference, where hidden states are produced in a single forward pass with teacher forcing on discrete tokens. The gradient attenuation we identify (Sec 3) arises specifically from this recurrent structure, not from model scale. This is precisely why process supervision is critical: it injects causal signals at each step of this recurrent chain, preventing the optimization from short-circuiting through the first latent position.
>
> This recurrent latent generation requires weight tying between embedding and output layers, since the hidden state from step t must serve as the input embedding for step t+1. Without weight tying, the output hidden states and input embeddings reside in misaligned representation spaces, making it hard for the model to coherently chain latent steps. This is why all foundational Latent CoT works use weight-tying architectures: COCONUT [1], CODI [2], and SemCoT [3] all adopt GPT-2. Our choice follows this established protocol.
>
> To directly address the scale concern, we conducted new experiments on **LLaMA-3.2-1B-Instruct** on GSM8K-Aug. Training setup: AdamW optimizer (LR=1e-4, WD=0.01), cosine schedule with warmup, gradient clipping 1.0, effective batch size 128, BF16 precision, 4x RTX 5880 Ada.
>
> | Method | Best Acc | Best Epoch |
> |--------|----------|------------|
> | Explicit CoT | **60.20%** | 10 |
> | OS-LATENT | 33.81% | 9 |
> | Answer Only | 32.98% | 8 |
>
> OS-LATENT (33.81%) collapses to the Answer Only baseline (32.98%), exactly replicating the optimization barrier observed with GPT-2. The latent tokens provide no benefit under outcome supervision, even at 8x scale on a modern architecture.
>
> **W2: Generalization and ULP Reliability**
>
> To further validate ULP, we conducted LM head projection and auxiliary decoder reconstruction experiments on latent states, showing that PS-GR states exactly reconstruct ground-truth reasoning steps **(see Reviewer N2XL Q4 for the full decode table)**.
>
> Beyond GSM8K, we extended to ProntoQA [4], a propositional logic QA benchmark requiring 3-6 hop deductive reasoning:
>
> | Method | ProntoQA Acc | ULP Probe Loss |
> |--------|-------------|----------------|
> | PS-LATENT | **97.40%** | **0.434** |
> | Explicit CoT | 80.80% | -- |
> | OS-LATENT | 78.60% | 0.546 |
> | No-CoT | 78.60% | -- |
>
> The same pattern holds on a structurally different task: OS-LATENT collapses to the No-CoT baseline, and ULP probe loss remains inversely correlated with accuracy (PS-LATENT 0.434 < OS-LATENT 0.546), confirming the Information Hierarchy generalizes beyond arithmetic.
>
> We also tested probe robustness by training three architecturally distinct probes (GPT-2, 2-layer MLP, 2-layer Transformer) on identical frozen hidden states. All three produce the identical ranking across all 6 model variants, confirming the hierarchy is a property of the representations, not probe design. **See Reviewer N2XL Q3 for the full comparison table.**
>
> We will incorporate all supplementary experiments (LLaMA-3.2-1B, ProntoQA, probe ablations, and decode analysis) into the revised manuscript. We sincerely appreciate the reviewer's insightful questions, which have helped strengthen the paper, and we kindly hope the reviewer will consider re-evaluating in light of the new evidence provided.
>
> [1] Hao et al., "Training LLMs to Reason in a Continuous Latent Space," COLM 2025
>
> [2] Shen et al., "CODI: Compressing CoT into Continuous Space via Self-Distillation," EMNLP 2025
>
> [3] He et al., "SemCoT: Accelerating CoT Reasoning through Semantically-Aligned Implicit Tokens," NeurIPS 2025
>
> [4] Saparov and He, "Language Models Are Greedy Reasoners," ICLR 2023

---

> > ### Author Rebuttal · Reviewer_sGK2 · 2026-04-03
> >
> > Thank you for the informative response. I have raised my rating.

---

> > > ### Author Response · Authors · 2026-04-04
> > >
> > > Thank you for re-evaluating our work and raising your rating. Your thoughtful questions helped us strengthen the paper considerably. All new results will be incorporated into the revised manuscript.

---

### Decision · Program_Chairs · 2026-04-30

**Decision:**

Accept (regular)

**Comment:**

This paper analyzes supervision in latent Chain-of-Thought through two orthogonal axes — Trajectory Control (Outcome vs Process Supervision) and State Alignment (Geometric Compression vs Generative Reconstruction). Experiments on GPT-2 / GSM8K-Aug argue that Outcome Supervision collapses to a No-CoT baseline, Process Supervision breaks that barrier, Geometric Compression acts as a destructive prior, and Generative Reconstruction preserves the reasoning manifold. A Unified Latent Probe reconstructing rationale steps from frozen latent states gives reconstruction losses that correlate with downstream accuracy across the $2 \times 2$ taxonomy.

**Strengths.** Reviewers agreed that the Trajectory Control $\times$ State Alignment decomposition is a useful conceptual framework for an otherwise fragmented latent-CoT literature (sGK2, N2XL, qbes). The central empirical claim — Outcome Supervision collapses while Process Supervision with Generative Reconstruction closes most of the gap to Explicit CoT — was materially strengthened during the rebuttal. The authors provided an exceptionally thorough response, including a LLaMA-3.2-1B scale replication, a ProntoQA cross-task replication, and a three-probe ablation, which successfully resolved reviewers' initial concerns regarding model scale and probe reliability.

**Weaknesses.** While the immediate methodological concerns were resolved, a broader limitation remains regarding empirical scope. The evaluation is primarily focused on structured or deductive reasoning tasks (GSM8K-Aug, ProntoQA). It remains an open question whether these information-theoretic dynamics and supervision strategies generalize equally well to more open-ended or higher-entropy reasoning settings such as long-context planning, code generation, or broader planning-style tasks.

**Decision.** The conceptual contribution is solid, and the authors' comprehensive rebuttal successfully fortified the evidence base, resolving the primary concerns of the engaged reviewers. Reviewer mfdY explicitly self-reported being unable to critically evaluate the paper; the weight of that review has therefore been heavily discounted. I recommend **accept**. The authors are strongly encouraged to incorporate the rebuttal experiments (LLaMA-3.2-1B, ProntoQA, probe ablations, and the decoding analysis) into the camera-ready version and to discuss the generalization boundaries of the framework more explicitly.